# Knowledge Distillation of BERT language model on the Arabic language

**Abrar Elidrisi, Hager Adil**
Department of Electrical and Electronics Engineering
University of Khartoum
{abrar.elidrisi, hageradil.ibr}@gmail.com

**Muhammed Saeed**
Universität des Saarlandes
musa00001@uni-saarland.de

**Tahani Attia**
Department of Electrical and Electronics Engineering
University of Khartoum
Tahani@uofk.edu

**Mohamed Saadeldin**
University of Dublin
Mohamed.saadeldin@ucd.iee

## Abstract

The absence of good Arabic language models led to significant setbacks in the Arabic language related tasks and lag with respect to robustness and accuracy. While a pre-trained version of BERT on Arabic language is available, a smaller distilled version could be proven to be highly scalable. In this research paper, we propose the development of DistilBERT for the Arabic language for the pursuit of achieving comparable results with significantly less computational resources. Employing knowledge distillation to create a compact model allows for wider implementation, even in areas with limited computational resources. Ultimately, this project aims to break down language barriers, bring greater inclusivity and improve the accessibility of the Arabic language in NLP applications worldwide. This project serves as a starting point for further research and investigation of the performance of the Arabic DistilBERT model across various NLP tasks.

## 1 Introduction

Despite the prevalence of the Arabic language worldwide, it remains one of the most underrepresented languages in the field of natural language processing (NLP), with a significant challenge posed due to its rich and complex morphology.
BERT language model (Bidirectional Encoder Representations from Transformers)(Devlin et al., 2018), a transformer-based architecture (Vaswani et al., 2017), has shown remarkable results in various NLP tasks. However, the high computational cost and large corpus required have limited its availability to high-resource languages. A pre-trained Arabic BERT model(Antoun et al., 2020) was developed to address the suboptimal performance achieved by the multilingual BERT for the Arabic language. Nonetheless, operating the model under resource constraints remains a challenge. To overcome this limitation, we propose employing the knowledge distillation technique to distill the knowledge learned by the large Arabic BERT into a smaller more efficient model, DistilBERT (Sanh et al., 2019) for the Arabic Language, while maintaining its performance. Evidently, our Arabic DistilBERT [1] [2] demonstrates a noticeable reduction in parameter count with only 108,871,680 parameters compared to the teacher model, Arabic-BERT-Large, which has 340,689,408 parameters. This reduction in size amounts to over 30%, a similar level of parameter reduction achieved in the English DistilBERT model. Additionally, we aim to maximize the performance of the Arabic DistilBERT for Question Answering [3] and identifying any challenges or limitations. Our ultimate goal is to facilitate inclusivity and accessibility of the Arabic language in NLP technologies thereby breaking down language barriers and promoting cross-cultural communication.

---

[1] https://huggingface.co/arabi-elidrisi/ArabicDistilBERT.

[2] https://github.com/ArabiElidrisi/Arabic_DistilBERT.

[3] https://huggingface.co/arabi-elidrisi/ArabicDistilBERT_QA.

## 2 METHOD

In this paper, We suggest developing an Arabic DistilBERT Model using the knowledge distilled from an Arabic BERT Model. Initially, we've introduced additional pre-processing to the data used. The dataset used was carefully selected to be public, large, relatively new with modern language, correct grammar and annotated correctly. As a result, we merged a few datasets, including OS-CAR 2019Suárez et al. (2019), Wikipedia Corpus, Arabic BERT Corpus and ARCD in text format. Aside from removing duplication, a pre-processing script was used to remove URLs, English words, emoticons, symbols flags, pictographs, transport and map symbols and unicodes. The results was a collection of over 90 million sentences of text. The goal was to create a balanced final version of the dataset with no duplication and suitable sentence length.

For the training stage, we employed the Knowledge Distillation technique, a compression technique to reproduce the behavior of the larger model into a smaller model using a triple loss function 2. We used a pre-trained Arabic BERT model with a general understanding of the context of the Arabic Language. The distilled model is then trained on the smaller corpus of the previously pre-processed Arabic dataset, leveraging on the knowledge transferred from the pre-trained Arabic BERT model.

$$L = \alpha * L_{distill} + \beta * L_{training} + \gamma * L_{cosine} \tag{1}$$

## 3 EXPERIMENTAL SETUP

We used Arabic BERT-large model as the teacher model. We initialized the DistilBERT model from the final weights of the teacher model. In addition, we modified its configurations to match those of the teacher, including the transformer layer's hidden state size set to $1024$, the attention heads to $16$, the intermediate size to $4096$ and vocabulary size to $32000$. The pre-training was done using 4 Tesla V100 GPUs, each of 32GB RAM. The pre-training process was carried out with a batch size of $16$, a learning rate of $3e-5$, and maximum sequence length of $512$ running on 3 epochs.

## 4 RESULTS

Being constrained to the resources available and the data size, the model training is currently ongoing. Therefore, we obtained a checkpoint upon the completion of the first epoch. The experimental results in Table 1 were obtained testing the performance for Question Answering on ARCD dataset. The Arabic DistilBERT achieved an F1 score of 62.20% comparable to 67.28% achieved by Arabic BERT-large model(Antoun et al., 2021)with over 30% reduction in parameter count. These findings provide evidence that the Arabic DistilBERT model can achieve competitive results on the Question Answering task while requiring less computational resources and training time.

Table 1: Comparison of the different models on the Question Answering task using ARCD

| $Model$ | $ARCD(F1)$ | $Parameters$ |
|---|---|---|
| Arabic DistilBERT | 62.20 | 108M |
| Arabic BERT-large | 67.28 | 340M |
| Arabic BERT-base | 62.24 | 110M |

## 5 CONCLUSION AND FUTURE WORK

We present Arabic DistilBERT, a language model that meets similar performance levels to its larger counterpart with significantly less computational resources. We showed that the model has the potential to notably contribute to the development of Arabic language-related tasks, improve its accuracy and robustness and thereby acts as a valuable addition to the natural language processing toolkit for the Arabic language. For future work, we aim to test the effectiveness of using the Arabic DistilBERT model across various NLP tasks.

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

## A  APPENDIX

Table 2: Examples from Arabic DistilBERT Model fine-tuned on Question Answering

| Context | Question | Answer | Score |
|---|---|---|---|
| عجائب الدنيا السبع الجديدة، هو مشروع استثماري أطلقه منتج أفلام كندي من أصل سويسري يدعى برنارد فير عام ١٩٩٩، وذلك من خلال شركة ربحية أنشأها باسم مؤسسة العالم المفتوح الجديد ( و اوبن وارلد ثربرت=ن). | من هو صاحب مشروع عجائب الدنيا السبع؟ | برنارد فير | 0.953 |
| على جانبي الحدود الثقافية بين أوروبا الجرمانية واللاتينية، وفي بلجيكا لغتين رئيسيتين هما: الهولندية المتكلمين بها (حوالي ٥٩ % )، ومعظمهم الفلمنكية، والمتكلمين بالفرنسية (حوالي ٤١ % )، ومعظمهم الولونيين، بالإضافة إلى مجموعة صغيرة من المتحدثين بالألمانية. اثنتين من أكبر المناطق في بلجيكا هي المنطقة الناطقة باللغة الهولندية الفلاندر في الشمال والمنطقة الناطقة بالفرنسية جنوب والونيا. في إقليم العاصمة بروكسل، ثنائية اللغة رسميا، هو جيب الناطقة بالفرنسية في الغالب داخل الإقليم الفلمنكي.* | بلجيكا لها لغتين رئيسيتين. اي من اللغتين يستخدمونه اكثر؟ | الهولندية | 0.836 |
| شهد حكم بشار الأسد بدأ موجة احتجاجات بدأت في ١٥ آذار عام ٢٠١١ ولا زالت رافعة شعارات ضد القمع والفساد وكبت الحريات ومطالبة بإسقاط النظام البعثي الذي استخدم ضدها الأسلحة الثقيلة وقوات الشبيحة بحسب منظميها في تحد غير مسبوق لحكم بشار الأسد وحزب البعث السوري ومتأثرة بموجة الاحتجاجات العارمة التي اندلعت في الوطن العربي أواخر عام ٢٠١٠، في حين أعلنت الحكومة السورية أن هذه الحوادث من تنفيذ متشددين وإرهابيين من شأنهم زعزعة الأمن القومي وإقامة إمارة إسلامية في بعض أجزاء البلاد. وقد نتج عن قمع هذا الحراك الشعبي الآلاف من الضحايا والجرحى والمعتقلين، بالإضافة إلى الخسائر في صفوف القوات المسلحة. | متى بدات الموجة الاحتجاجية في عهد بشار؟ | ١٥ آذار عام ٢٠١١ | 0.958 |

