# OpenReview forum: "Knowledge Distillation of BERT Language Model on the Arabic Language"
_ICLR.cc/2023/TinyPapers — Submitted to Tiny Papers @ ICLR 2023_

### Official Review · Reviewer_k8vi · 2023-03-28

**Confidence:** 5

**Summary Of Contributions:**

The development of a DistilBERT for the Arabic language is proposed. The proposed model has also been tested in a QA task.

**Rating:**

Clear, Correct, and Reproducible (CCR): a submission which meets the reviewing criteria

**Strengths And Weaknesses:**

Arabic DistilBERT is presented. For low-resource languages having such models is a great asset.

The manuscript is written well and meets the basic requirements of the “Tiny” paper. The authors claim that Arabic distilBERT will require fewer resources to achieve comparable performance as Arabic BERT. A table or some comparison of resources would have helped justify the “fewer” resources requirement. The paper expects the readers to take the author's word for such a claim.

The results could be presented better. The argument for needing Arabic DistilBERT is simple - a model with fewer resources that performs equally well as the existing BERT. So the results should show that. A statistical comparison between model resources and model performances might be more suitable here.

Also, will the Arabic DistilBERT be available for other researchers to access and use? (i.e. maybe at huggingface)


**Suggested Changes:**

Results need to be presented better. I have included suggestions in the above review.

---

### Author Response · Authors · 2023-06-11
**A response to the reviewers suggestions on the paper**

Thank you for the time and effort that you have committed to carefully reviewing our paper and generously providing us with valuable feedback which we have taken into consideration and made the necessary modifications.

1- To guarantee proper acknowledgment of the dataset used, we have included a reference to the source of the Oscar dataset in our manuscript.

2- We understand the requirement for clarity about statistical comparison between the Arabic DistilBERT model and Arabic BERT-Large model to further support the argument. As suggested, we have elaborated on this in the revised manuscript. As compared to its teacher model, Arabic-BERT-Large, which incorporates 340,689,408 parameters, the Arabic DistilBERT model maintains over 30% of the parameters, amounting to 108,871,680 parameters. This represents a significant reduction in size and offers a lighter model architecture.

3- Additionally, we have added a new table column in the results section, listing the number of parameters for each model. This should provide a clearer understanding of the comparison between different models.

4- We have made our model open-source and freely available for the research community. The model can be accessed from GitHub at this link:
https://github.com/ArabiElidrisi/Arabic_DistilBERT
And also available in Hugging face at:
https://huggingface.co/arabi-elidrisi/ArabicDistilBERT

5- We have also made our Arabic DistilBERT model fine-tuned on Question Answering task available at :
https://huggingface.co/arabi-elidrisi/ArabicDistilBERT_QA

We express our gratitude to the reviewers for their willingness to consider our work and for their valuable feedback. We look forward to hearing any additional comments or suggestions you may have to further enhance the quality of our manuscript.

We extend our sincere appreciation to the esteemed reviewers for  for their willingness to consider and evaluate our work and providing valuable feedback. We look forward to receiving any further comments or suggestions that they may have, as we believe it will greatly contribute to enhancing the overall quality of our manuscript.

---

> ### Comment · Area_Chair_rB5x · 2023-06-15
> **Thank you for addressing the comments of the reviewers**
>
> Dear Authors,
>
> We greatly appreciate you taking the time and effort to address the critical issues raised by the reviewers.
>
> We are glad to learn that you learned from the reviews and I'm really impressed by the details and how the content of the work has improved.
>
>
> Please feel free to let us know if you have other questions or if you want any feedback.
>
> Thank you!!

---

### Meta-Review · Area_Chair_rB5x · 2023-04-06

**Recommendation:** Invite to archive
**Confidence:** 4

**Metareview:**

**Summary**
* The paper introduces a DistilBERT for the Arabic language where there are not previous simple BERT versions for the language. The authors tested the proposed method is tested in a Q&A task.

**Strength**
* Developing a distilled version of BERT for the Arabic language is important.

**Weakness**
* The dataset used for training is not shared. Since it’s a public dataset, it would be great to add it for reproducibility purposes.
A statistical comparison of this distilled BERT and the original Arabic BERT might be useful.


**Summary:**

The paper introduces a DistilBERT for the Arabic language where there are not previous simple BERT versions for the language. The authors tested the proposed method is tested in a Q&A task. Strength: developing a distilled version of BERT for Arabic language is important. Weakness: The dataset used for training is not shared. It would be great to add it for reproducibility purposes.

**Comments And Feedback To The Authors:**

- Motivation is great and important. However, there's limited discussion about the dataset and the model. Please include these details and share the model.

**Reason For Not Giving A Higher Recommendation:**

* The paper would greatly benefit by adding more information about the data used for training and sharing the model. The authors spent much of the paper in the introduction; they could use the space to discuss the necessary details.
* Mainly, at its current state this paper doesn't fulfill the reproducibility criteria of Tiny Papers.

**Reason For Not Giving A Lower Recommendation:**

N/A

---

### Decision · Program_Chairs · 2023-04-07

Invite to archive